# Examining Factors Associated with Dynapenia/Sarcopenia in Patients with Schizophrenia: A Pilot Case-Control Study

**DOI:** 10.3390/healthcare11050684

**Published:** 2023-02-25

**Authors:** Ryuichi Tanioka, Kyoko Osaka, Hirokazu Ito, Yueren Zhao, Masahito Tomotake, Kensaku Takase, Tetsuya Tanioka

**Affiliations:** 1Faculty of Health Sciences, Hiroshima Cosmopolitan University, Hiroshima 731-3166, Japan; 2Department of Nursing, Nursing Course of Kochi Medical School, Kochi University, Kochi 783-8505, Japan; 3Graduate School of Biomedical Sciences, Tokushima University, Tokushima 770-8509, Japan; 4Department of Psychiatry, Fujita Health University, Toyoake 470-1192, Japan; 5Department of Rehabilitation, Anan Medical Center, Anan 774-0045, Japan

**Keywords:** dynapenia, sarcopenia, schizophrenia, basal metabolic rate, total muscle mass, skeletal muscle mass, body mass index

## Abstract

Sedentary behavior in patients with schizophrenia causes muscle weakness, is associated with a higher risk of metabolic syndrome, and contributes to mortality risk. This pilot case-control study aims to examine the associated factors for dynapenia/sarcopenia in patients with schizophrenia. The participants were 30 healthy individuals (healthy group) and 30 patients with schizophrenia (patient group), who were matched for age and sex. Descriptive statistics, Welch’s *t*-test, cross-tabulations, adjusted residuals, Fisher’s exact probability test (extended), and/or odds ratios (ORs) were calculated. In this study, dynapenia was significantly more prevalent in patients with schizophrenia than in healthy individuals. Regarding body water, Pearson’s chi-square value was 4.41 (*p* = 0.04), and significantly more patients with dynapenia were below the normal range. In particular, body water and dynapenia showed a significant association, with an OR = 3.42 and 95% confidence interval [1.06, 11.09]. Notably, compared with participants of the healthy group, patients with schizophrenia were overweight, had less body water, and were at a higher risk for dynapenia. The impedance method and the digital grip dynamometer used in this study were simple and useful tools for evaluating muscle quality. To improve health conditions for patients with schizophrenia, additional attention should be paid to muscle weakness, nutritional status, and physical rehabilitation.

## 1. Introduction

The prevalence of schizophrenia in Japan is estimated at 0.7% [1]. Patients with schizophrenia die on average 10 to 20 years earlier than healthy individuals [2,3]. The sedentary lifestyle common among patients with schizophrenia is associated with higher metabolic syndrome (cardiovascular changes due to diabetes mellitus, hypertension, and hypercholesterolemia) and contributes to mortality risk. Lifestyles that increase the risk of such metabolic syndrome have been identified as lacking regular physical activity, poor food intake, substance use, and high rates of smoking [4].

Strassnig et al. [5] developed a comprehensive model to conceptualize multimodal relationships that predict impaired activities of daily living in patients with schizophrenia. According to these authors, limitations in physical abilities interfere with activities of daily living and elicit a state of physical infirmity observed in other chronic illnesses. A high prevalence of sarcopenic obesity has also been reported in patients with schizophrenia [6]. However, little is known on the risk factors for dynapenia/sarcopenia in patients with schizophrenia.

Factors that contribute to severe limitations due to the pathophysiology of schizophrenia and the effects of medications are complex and include a sedentary lifestyle as well as factors due to the effects of medication therapy, which ultimately lead to a vicious cycle of obesity and cardiovascular metabolic risk [7]. A sedentary lifestyle and decreased functional motor skills in patients with schizophrenia reduce their quality of life [8]. Low physical activity levels are also associated with the use of antipsychotic drugs. This implies that increasing weight is related to limitations in physical functioning and restricts activities of daily living. Physical inactivity due to obesity also adds to the burden of schizophrenia in the form of reduced physical health-related quality of life. Rehabilitation programs focusing on these risk factors should be key for physical activity for both prevention and treatment of disease and disablement in patients with schizophrenia [9]. Moreover, antipsychotic medications have been associated with weight gain and obesity in schizophrenia. Patients with schizophrenia consume unhealthy food, and their dietary patterns identified a high consumption of saturated fat and low intake of fruit and dietary fiber [10]. In patients with schizophrenia, the ability to supply oxygen to muscles during exercise and the ability of muscles to consume oxygen (cardiopulmonary endurance) are poorer than in healthy individuals. This means the level of cardiorespiratory fitness may be extremely low in patients with schizophrenia, amounting to a state of deconditioning and a very low capacity for sustained physical activity that is high in intensity, activities promoting low to moderate activity levels may serve the population well and lead to highly relevant improvements in health prospects [11].

In a 12-year follow-up study of schizophrenia, the patient group had an excess of psychiatric and physical comorbidities (fractured neck of femur, parkinsonism, pneumonia, esophageal ulcer, respiratory failure, and bronchitis), including side effects of psychotropic drugs, compared to the age- and sex-matched controls. Specifically, their finding clearly demonstrates that parkinsonism-associated complications may play a dominant role in schizophrenia-related death in general hospitals. Reducing the risk of parkinsonism-associated complications due to accurate detection and management of side effects of psychotropic and somatic medication as well as of related drug–drug interactions, continuously monitoring physical status, and accurate detection of concomitant metabolic, cardiovascular, and respiratory diseases as well as creating awareness about preventive strategies for difficulty eating and aspiration pneumonia may help in reducing parkinsonism-associated fatal consequences in general hospitals in patients with schizophrenia [12].

In the context of socioeconomic challenges, schizophrenia leads to particularly unhealthy lifestyles that include poor diets, little exercise, marked sedentary behavior, and high rates of smoking with commensurately low physical activity levels [13]. As a result, compared to the general healthy population, patients with schizophrenia have severe symptomatic limitations in physical capacity. Negative symptoms reduce the likelihood of patients’ engagement in goal-directed behavior, including physical activity, which has been noted to increase obesity and cardiometabolic risk and induce poor physical conditions, resulting in sarcopenic obesity and muscle weakness [6].

According to the European working group on sarcopenia in older people 2 (EWGSOP 2), sarcopenia is suspected when (1) muscle weakness is confirmed, whereas sarcopenia is confirmed when (2) muscle mass or muscle quality decline is present in addition to muscle weakness [14]. As mentioned above, sarcopenia is defined as “loss of muscle mass or quality,” whereas dynapenia is defined as “loss of muscle strength” [15].

Sarcopenia is also associated with depressed mood, which in turn is associated with low muscle strength and physical performance [16]. Therefore, it is problematic that patients with schizophrenia frequently have negative symptoms such as depressed mood, which is associated with low physical function and low muscle strength. Regarding schizophrenia and nutritional status, Japanese inpatients with schizophrenia are more likely to be underweight and undernourished than outpatients [17]. Nutritional status is an issue for patients with schizophrenia in Japan. Therefore, it is important to consider the activities of daily living, dynapenia, sarcopenia/presarcopenia, and nutritional status when considering symptom management in hospitalized patients with chronic mental illness.

This pilot study aimed to examine the associated factors for dynapenia/sarcopenia in patients with schizophrenia.

## 2. Materials and Methods

### 2.1. Study Participants

This pilot case-control study enrolled 60 individuals in total comprising 30 healthy participants (healthy group) and 30 patients with schizophrenia (patient group), matched by age, ranging from 40 to 89 years, and sex.

### 2.2. Data Acquisition Period

The study’s data acquisition phase was from 17 August 2021 to 30 November 2021.

### 2.3. Target Selection Criteria

Healthy group: Employees working at Hospital A and its Geriatric Health Care Facility.

Patient group: Patients with schizophrenia admitted to Hospital A.

Both groups were matched by sex and age.

### 2.4. Exclusion Criteria

Healthy group: Individuals with a mental or physical disorder.

Patient group: Individuals unable to understand instructions owing to a medical condition or medication status or because of a physical disorder such as a history of cerebrovascular disease such as stroke or a neurological disease.

### 2.5. Assessment Methods

#### 2.5.1. Body Mass

Tanita monitors use the latest bioelectrical impedance analysis technology, first developed by Tanita in 1992, to provide fast and accurate body composition results [18]. The RD-545 InnerScan Pro provides an in-depth analysis of 26 body composition measurements. The measurements included weight, body fat, muscle mass, muscle quality score, body mass rating, bone mass, visceral fat level, basal metabolic rate, metabolic age, total body water, and body mass index (BMI). The RD-545 InnerScan PRO can perform fat and muscle analysis individually for arm, leg, and trunk segments if hand electrodes are used [19]. The state of visceral fat accumulation is indicated as visceral fat level score measured by the RD-545 InnerScan Pro.

#### 2.5.2. Age, Height, and Weight

Healthy group: Age and height were self-reported based on the hospital’s staff health examination form.

Patient group: Age and height were obtained from the medical records.

Weight was measured in both groups using a scale (RD-545 InnerScan Pro, TANITA Corporation. Tokyo, Japan).

#### 2.5.3. Grip Strength of the Hands

A digital grip dynamometer (T.K.K.5401; Takei Scientific Instruments, Co., Ltd., Niigata, Japan) was used to individually measure the grip strength of each hand in a stable standing posture.

#### 2.5.4. Skeletal Muscle Mass Index (SMI)

The total limb skeletal muscle mass (kg) was calculated from the information obtained from the body mass, and the data were divided by the square of the corresponding height (m^2^).

#### 2.5.5. SARC-F Score

The SARC-F score was presented by Morley as a screening tool for sarcopenia at the EU/US committee on sarcopenia in the frail elderly at the Conference on Sarcopenia Research (ICSR) in Orlando in 2012 [20].

Data were self-reported by all participants using a questionnaire survey.

### 2.6. Sarcopenia/Dynapenia Assessment Method

This study adopted the diagnostic criteria proposed by the Asian Working Group for Sarcopenia (AWGS) [21]. The SARC-F score, grip strength, and skeletal muscle mass were used as indicators. The specific criteria were as follows:

Grip strength can be used to assess muscle weakness [20,21]. Peripheral quantitative computed tomography, dual X-ray energy absorptiometry, and magnetic resonance imaging techniques can be used to assess skeletal muscle mass and quality [22]. Other than the aforementioned methods, bioelectrical impedance analysis can be used, which has the advantage of being inexpensive and portable. The cutoff values for sarcopenia in the Japanese population are 6.8 kg/m^2^ for men and 5.7 kg/m^2^ for women [23].

The SARC-F score was used to select participants with sarcopenia; those with a score of 4 or more points were selected. Based on the two sarcopenia criteria outlined in the Section 1, muscle weakness and loss of muscle mass or muscle quality were evaluated. (1) Grip strength was used as an indicator of muscle weakness, defined for men and women as having a grip strength of less than 26 kg and less than 18 kg, respectively. In addition, (2) skeletal muscle mass (kg/m^2^) was used as an indicator of muscle mass or muscle quality loss, and skeletal muscle mass loss was defined as a value less than 7.0 kg/m^2^ for men and less than 5.7 kg/m^2^ for women. Presarcopenia was defined as reduced skeletal muscle mass and normal grip strength. Dynapenia was defined as a normal skeletal muscle mass and decreased grip strength.

### 2.7. Statistical Analysis

Basic statistical parameters (mean ± standard deviation [SD], 95% confidence interval [CI]) were calculated. Welch’s *t*-test was performed to compare the two study groups. For items that were significantly different between the two groups, cross-tabulations were performed, and adjusted residuals were calculated. Fisher’s exact probability test (Extended), Pearson’s chi-square test, and/or odds ratios (ORs) were calculated. All statistical analyses were performed using SPSS 21.0 (IBM Corporation). Statistical significance was set at *p* < 0.05.

## 3. Results

Among the study participants, 61.7% (37/60) were women and 38.8% (23/60) were men. The healthy group comprised 63.3% (19/30) women and 36.7% (11/30) men, whereas the patient group consisted of 60.0% (18/30) women and 40.0% (12/30) men.

In this study, dynapenia and sarcopenia/presarcopenia were assessed. Among the 30 participants in the patient group, 10.0% (3/30 [1/18 women, 2/12 men]) met the criteria for sarcopenia, 3.3% (1/30 [1/12 men]) for presarcopenia, and 60.0% (18/30 [14/18 women, 4/12 men]) for dynapenia. The corresponding results of the healthy group showed that sarcopenia and presarcopenia were not present (0%) and that 13.3% (4/30 [3/19 women, 1/11 men]) met the criteria for dynapenia.

Table 1 shows the results of Welch’s *t*-test. Body water content was significantly higher in the healthy group with 53.56 ± 3.94% in the healthy group and 49.77 ± 6.58% in the patient group (*t* = 2.71, *p* < 0.001).

The visceral fat level score was 6.60 ± 3.71 in the healthy group and 9.12 ± 5.35 in the patient group (*t* = 2.11, *p* = 0.04). Body fat content was 24.95 ± 6.05% in the healthy group and 30.41 ± 9.00% in the patient group (*t* = 2.76, *p* < 0.01). Likewise, BMI was 21.89 ± 2.30 kg/m^2^ for the healthy group and 23.88 ± 4.65 kg/m^2^ for the patient group (*t* = 2.10, *p* = 0.04).

Left grip strength was 29.16 ± 9.07 kg for the healthy group and 18.53 ± 8.38 kg for the patient group (*t* = 4.71, *p* < 0.001), whereas right grip strength was 30.05 ± 7.98 kg for the healthy group and 21.26 ± 10.92 kg for the patient group (*t* = 3.56, *p* < 0.001). These findings showed that for both sides, the grip strength of the patient group was significantly weaker than that in the healthy group.

As shown in Table 2, the patient group was significantly more likely to have dynapenia or sarcopenia/presarcopenia (Fisher’s exact test, *p* < 0.0001; OR, 17.88; 95% CI [4.74, 67.43]).

The association of the study group with dynapenia, including sarcopenia and presarcopenia (hereafter referred to as dynapenia in the Section 3), was analyzed based on items with significant differences in Table 1. No significant association was found for the parameters of visceral fat level score, body fat, and BMI.

In contrast, for body water, the result of Pearson’s chi-square test was 4.41 (*p* = 0.04), and significantly more people with dynapenia were below the normal range. We also confirmed a significant association for body water (OR, 3.42, 95% CI [1.06, 11.09]).

## 4. Discussion

As shown in Table 1, the patient group had significantly higher body fat, visceral fat level scores, and BMI. In addition, the average value of the patient group BMI is not at the obese level, and the high visceral fat level score was deemed a problem when considered overall from the cross-tabulation results in Table 2.

As shown in Table 2, the patient group had a high percentage of individuals diagnosed with dynapenia, with an OR of 17.88 times the risk of developing the disease compared with healthy individuals. Thus, it was suggested that being afflicted with schizophrenia is one factor associated with dynapenia. Moreover, Table 2 shows that no significant association by the study group was found for body fat, visceral fat level score, or BMI; however, body water content was significantly associated, with the OR indicating 3.42 times higher risk of dynapenia for the patient group than for the healthy group.

For these reasons, the patient group in this study may have increased fat, as well as decreased body water content and muscle mass, owing to a sedentary lifestyle [9,24]. Sex differences in body fat and water content in patients with schizophrenia have been reported [6]. The body water content was predominantly higher in the healthy group. Body water refers to water contained in various body compartments, including blood, lymphatic fluid, extracellular fluid, and intracellular fluid [25]. These fluids play important roles in the body, such as transporting nutrients and maintaining a constant body temperature, and they tend to decrease with age. In addition, people with high body fat tend to have a lower body water content [26]. This trend is also consistent with the previous study by Bulbul et al. [27] Therefore, it is necessary to focus on the trends of high body fat and low water content in the patient group.

Of the 307 participants in the study by Mori et al. [28], 60.9% were assessed as normal, and 25.7%, 8.1%, and 5.2% were found to have presarcopenia, sarcopenia, and dynapenia, respectively. Reduced grip strength is a critical indicator of dynapenia [29]. In this study, grip strength was significantly lower in patients with schizophrenia than in healthy individuals. Because many patients with schizophrenia have dynapenia, grip strength may be a convenient screening index for dynapenia in psychiatric hospitals.

The participants of the study by Kobayashi et al. were volunteers aged over 60 years who were in good general health [30]. Their study found that in Japan, the rates of sarcopenia, presarcopenia, and dynapenia were 10%, 22%, and 8% in men, and 19%, 23%, and 13% in women.

According to Neves et al., sarcopenia and dynapenia were identified in 15.3% and 38.2% of old persons [31]. In this study, 13.3% of the healthy individuals had dynapenia, whereas 60.0% of the patient group had dynapenia, 10.0% had sarcopenia, and 3.3% had presarcopenia. Thus, our data suggest that the prevalence of dynapenia is high among patients with schizophrenia.

Appetite regulation and physical activity affect energy balance and changes in body fat mass. In some patients, inflammation induces anorexia and fat loss along with sarcopenia. In others, appetite is maintained, despite the activation of systemic inflammation, leading to sarcopenia with normal or increased BMI. Inactivity contributes to sarcopenia and increased fat tissue in aging and disease [32].

In a previous study of the BMI status of hospitalized Japanese schizophrenia patients, underweight and obesity were characteristic in schizophrenia inpatients compared with the general population. In particular, regarding the characteristics of underweight, a previous study showed that the prevalence of hypotriglyceridemia was significantly higher in the underweight group than in the normal weight group and in overweight/obese schizophrenia inpatients [33].

Harvey and Strassnig [34] suggested that the cognitive limitations of people with schizophrenia not only correlate with disability directly, but contribute substantially to other skills deficits (functional capacity; social competence) that exacerbate disability outcomes. Impaired cognition and negative symptoms, particularly in the domains of reasoning and problem solving and reinforcement valuation, can lead to deficits in functional capacity that then lead to poor dietary and exercise choices, contributing to poor functional outcomes.

In another study, age, certification of long-term care, and malnutrition were identified as risk factors for sarcopenia [35]. Sarcopenia is thought to primarily explain the age-related loss of muscle strength, such as dynapenia, commonly seen in older people [36]. However, recent longitudinal data indicate that the loss of muscle strength occurs significantly faster than the accompanying loss of muscle mass [37]. On the other hand, gains in muscle mass and strength afforded by resistive training are associated with a small but significant improvement in physical performance. It is noteworthy that lower intensity mechanical loading such as aerobic exercise, despite being considerably less effective for inducing muscle hypertrophy, has been found to promote protein synthesis and expression of growth-related genes and inhibit the expression of muscle breakdown-related genes [37].

Muscle weakness is known to decrease physical function and increase the risk of mortality [38]. Regarding the changes in physical function associated with aging, muscle strength declines by 30% and muscle area by 40% between 20 and 70 years of age [39]. At the age of 75 years, muscle strength declines at a rate of 2.5–3% per year for women and 3–4% per year for men, and muscle mass is lost at a rate of 0.64–0.70% per year for women and 0.80–0.98% per year for men [37]. Kitamura, et al. [40] found sex-specific patterns of correlates with sarcopenia. Significant sarcopenia-related factors in addition to ageing were hypoalbuminaemia, cognitive impairment, low activity, and recent hospitalization among men and cognitive impairment and depressed mood among women. It is important to focus on these conditions.

Compared to young adults, older adults have a lower limb skeletal muscle index (ASMI, kg/m^2^) and a significantly higher body fat percentage [41]. It has been noted that diabetic patients with a high body fat percentage in addition to low BMI may develop sarcopenia [42]. Moreover, the prevalence of diabetes in patients with schizophrenia in Japan has been reported to be 8.6% [43].

Protein intake is necessary for efficient muscle growth. A person with adequate muscle mass needs 1.0–1.2 g protein per kg of body weight per day for an older person to maintain muscle mass, i.e., about 60–72 g per day if the person weighs 60 kg [44]. However, this intake is not sufficient for those who must gain muscle mass due to sarcopenia, and they should have an intake of 1.2–1.5 g of protein per kg of body weight per day, i.e., 72–90 g per day if they weigh 60 kg [45]. Thus, it is important to control the balance of restricted caloric intake with guaranteed protein intake for patients with dynapenia. However, if a patient has kidney problems, it is critical to pay much more attention to an appropriate protein and calorie intake during the rehabilitation process [46].

Based on the BMI findings of our study, the patient group was not underweight. Our study subjects were inpatients; they have consumed a diet regulated by a psychiatrist and a dietitian. However, outpatients may not be eating an appropriate diet due to unbalanced diets, poverty, etc.

With this in mind, we should conduct the main case-control study following this pilot study. Furthermore, inpatients may have a lower average BMI than outpatients, who are free to eat whatever they want at home because their food intake is controlled to prevent excessive weight gain. It was considered important to keep these points in mind when managing their health.

### Limitations and Future Research

Since data on daily intakes, such as nutritional status, were not obtained in this study, it is necessary to obtain data on “official” caloric intake based on hospital diets, such as daily caloric intake, for better analyses in future studies. Additionally, it is necessary to consider “unofficial” caloric intake, such as snacks. Moreover, the patient’s amount of activity needs to be considered. This pilot study was a small-scale study conducted to inform, predict, and direct an intended future full-scale study. The association of low body water and dynapenia in patient participants suggests that low body water might be a risk factor for dynapenia in these patients.

Underweight is highly prevalent in Japanese inpatients with schizophrenia. Psychiatrists should be aware of underweight and their potential health risks. Treating psychiatrists should also be responsible for providing any necessary nutritional interventions [47]. Physical health appears to be achievable in people with schizophrenia being challenged by motivational difficulties with attending regular exercise and have beneficial implications for physical function during activities of daily living, lifestyle-related diseases, and early death. Specifically, physical training is an effective countermeasure to improve the low aerobic endurance and skeletal muscle strength in these patients [48].

Furthermore, the main study following this pilot study should include not only body composition (low body water, visceral fat level, and muscle mass), grip strength, and joint range of motion, but also medication content, heart rate variability, and motor velocity [49]. Other factors (physical function during activities of daily living, gait and psychiatric symptoms specific to schizophrenia, age, and length of hospitalization) also must be considered in dynapenia in patients with schizophrenia.

## 5. Conclusions

This pilot study examined the risk factors for dynapenia/sarcopenia in patients with schizophrenia. Patients with schizophrenia were overweight, had less body water than the healthy study participants, and were at a higher risk of dynapenia than participants in the healthy group. The impedance method used in this study is a simple and useful method for evaluating muscle quality in conditions such as dynapenia. To improve health conditions for patients with schizophrenia, additional attention should be paid to muscle weakness, nutritional status, and physical rehabilitation. Future research will include a larger study following on this pilot study.

## Figures and Tables

**Table 1 healthcare-11-00684-t001:** Comparison of the measured parameters between the two study groups.

Items	Healthy Group		Patient Group		Welch’s *t*-Test		95% Confidence Interval	
	*n* = 30		*n* = 30					
	Mean	±SD	Mean	±SD	*t*	*p*	Lower	Upper
Age	64.3	10.16	62.53	10.26	0.67	0.51	−3.51	7.04
Body height, cm	161.47	6.11	159.5	9.08	0.98	0.33	−2.04	5.97
Body weight, kg	57.25	8.49	60.55	15.52	−1.03	0.31	−9.82	3.2
BMR, kcal	1195.9	176.66	1216.97	276.33	−0.35	0.73	−141.38	99.25
Bone mass, kg	2.37	0.34	2.25	0.46	1.12	0.27	−0.09	0.33
Body water, %	53.56	3.94	49.77	6.58	2.71	<0.01	0.98	6.61
Visceral fat level score	6.6	3.71	9.12	5.35	−2.11	0.04	−4.91	−0.13
Body fat, %	24.95	6.05	30.41	9	−2.76	<0.01	−9.44	−1.49
BMI, kg/m^2^	21.89	2.3	23.88	4.65	−2.1	0.04	−3.91	−0.08
MM L-arm, kg	2	0.45	1.99	0.63	0.11	0.92	−0.27	0.3
MM R-arm, kg	2.02	0.46	1.88	0.54	1.09	0.28	−0.12	0.4
MM upper limb, kg	4.02	0.9	3.87	1.11	0.59	0.56	−0.37	0.68
MM L-leg, kg	7.49	1.26	7.01	1.75	1.2	0.24	−0.32	1.26
MM R-leg, kg	7.55	1.27	6.99	1.7	1.43	0.16	−0.22	1.33
MM lower limb, kg	15.04	2.52	14.01	3.4	1.33	0.19	−0.52	2.57
MM trunk, kg	21.48	3.48	21.63	4.42	−0.14	0.89	−2.21	1.91
TMM, kg	40.54	6.72	39.51	8.67	0.52	0.61	−2.98	5.04
LGS, kg	29.16	9.07	18.53	8.38	4.71	<0.001	6.12	15.15
RGS, kg	30.05	7.98	21.26	10.92	3.56	<0.001	3.83	13.74
SMI, kg/m^2^	7.26	0.82	6.94	1.18	1.21	0.23	−0.21	0.84

SD, standard deviation; BMR, basal metabolic rate; BMI, body mass index; MM, muscle mass; L-arm, left arm; R-arm, right arm; L-leg, left leg; R-leg, right leg; TMM, total muscle mass; LGS, left grip strength; RGS, right grip strength; SMI, skeletal muscle mass index.

**Table 2 healthcare-11-00684-t002:** Associations with the risk of dynapenia/sarcopenia in healthy individuals and patients with schizophrenia.

Group		Healthy	Schizophrenia	Analysis Results	
Normal	Frequency	26	8	Fisher’s exact test, *p* < 0.0001, OR = 17.88, 95% CI [4.74, 67.43]	
	AR	4.7	−4.7		
Dynapenia *	Frequency	4	22		
	AR	−4.7	4.7		
**Body Water**		**Normal**	**Below Standard**	Pearson’s chi-square test = 4.413, *p* = 0.04, OR = 3.42, 95% CI [1.06, 11.09]	
Normal	Frequency	28	6		
	AR	2.1	−2.1		
Dynapenia *	Frequency	15	11		
	AR	−2.1	2.1		
**Visceral Fat Level Score**		**Normal**	**Above Standard**	Pearson’s chi-square test = 0.184, *p* = 0.67, OR = 1.27, 95% CI [0.43, 3.80]	
Normal	Frequency	24	10		
	AR	0.4	−0.4		
Dynapenia *	Frequency	17	9		
	AR	−0.4	0.4		
**Body Fat**		**Decreased**	**Normal**	**Increased**	**Analysis Results**
Normal	Frequency	6	20	8	Fisher’s exact test, *p* = 0.159
	AR	0.2	1.6	−1.8	
Dynapenia *	Frequency	4	10	12	
	AR	−0.2	−1.6	1.8	
**BMI Level**		**Decreased**	**Normal**	**Increased**	**Analysis Results**
Normal	Frequency	2	24	8	Fisher’s exact test, *p* = 0.774
	AR	−0.8	0.7	−0.3	
Dynapenia *	Frequency	3	16	7	
	AR	0.8	−0.7	0.3	

* Dynapenia includes dynapenia (*n* = 22), presarcopenia (*n* = 1), and sarcopenia (*n* = 3). AR, adjusted residual; BMI, body mass index; CI, confidence interval; OR, odds ratio.

## Data Availability

Data presented in this study are available upon request from the corresponding author. The data are not publicly available because of privacy and ethical restrictions.

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
