# Peer review of "Examining Factors Associated with Dynapenia/Sarcopenia in Patients with Schizophrenia: A Pilot Case-Control Study"

_healthcare, 2023, doi:10.3390/healthcare11050684_

Round 1

Reviewer 1 Report

Thank you for conducting this critical study. Few items that need clarifications:

Lines 60-64- please clarify this statement, "patients with schizophrenia have severe limitations in physical capacity due to aging and sedentary lifestyle caused by patients with schizophrenia (?)

Line 68, page 2 of 10, it seemed that this statement is missing something, in addition to(?)

Lines 74 and 75 would strengthen the argument if authors could include why Japanese inpatients with schizophrenia are more likely to be underweight and undernourished than outpatients.

Under 2. Materials and Methods- Please indicate the study design, i.e., case-control study.

Under 3. Results, lines 163 to 175, it would be better if differences between male and female (or by sex) were also presented since study participants were matched by age and sex (lines 85 and 86), or at least include it in the table. For age, I see it was used as a continuous variable. Do you think it would be good to categorize age at least into two categories (adults and older adults (65 years old and above) to examine variations in muscle mass?

Lines 157-162 states that 10% had sarcopenia (sounds diagnostic, conclusive). How about stating that 10% met the criteria for sarcopenia since no physical performance was measured as part of the assessment of sarcopenia?

Author Response

Thank you for conducting this critical study. Few items that need clarifications:

Response: Thanks for your comments, we appreciate it. Revisions are shown in red font.

Lines 60-64- please clarify this statement, "patients with schizophrenia have severe limitations in physical capacity due to aging and sedentary lifestyle caused by patients with schizophrenia (?)

Response: Thanks for your advice. The following modifications have been made and references were added as follows.

In the context of socioeconomic challenges, schizophrenia leads to particularly unhealthy lifestyles that include poor diets, little exercise, marked sedentary behavior, and high rates of smoking with commensurately low physical activity levels [12]. As a result, compared to the general healthy population, patients with schizophrenia have severe symptomatic limitations in physical capacity. Negative symptoms reduce the likelihood of patients’ engagement in goal-directed behavior, including physical activity, which has been noted to increase obesity and cardiometabolic risk and induce poor physical conditions, resulting in sarcopenic obesity and muscle weakness [5].

Line 68, page 2 of 10, it seemed that this statement is missing something, in addition to(?)

 Response: We have revised like below.

According to the European Working Group on Sarcopenia in Older People 2 (EWGSOP 2), sarcopenia is suspected when (1) muscle weakness is confirmed, whereas sarcopenia is confirmed when (2) muscle mass or muscle quality decline is met in addition to muscle weakness [13].

Lines 74 and 75 would strengthen the argument if authors could include why Japanese inpatients with schizophrenia are more likely to be underweight and undernourished than outpatients.

Response: Visceral fat level score (t = 2.11, p = 0.04), body fat (t = 2.76, p <0.01), and BMI (t = 2.11, p = 0.04) were significantly higher in the patient group than in the healthy group. Thus, visceral fat level score, body fat, and BMI were significantly higher in the patient group than in the healthy group. However, the average value of the patient group in BMI is not at the obese level, and the high visceral fat level score was considered a problem when considered overall from the cross tabulation results in Table 2.

Therefore, we have revised the discussion sentences like below:

As shown in Table 1, the patient group had significantly higher body fat, visceral fat level scores, and BMI. However, it was suggested that being afflicted with schizophrenia is one factor associated with dynapenia. Also, the average value of the patient group BMI is not at the obese level, and the high visceral fat level score was deemed a problem when considered overall from the cross-tabulation results in Table 2.

Additionally, the following sentences have were added to the last part of the discussion section.

Based on the BMI findings of our study, the patient group was not underweight. Our study subjects were inpatients, they have consumed a diet regulated by a psychiatrist and a dietitian. However, outpatients may not be eating an appropriate diet due to unbalanced diets, poverty, etc.

With this in mind, we should conduct the main case-control study following this pilot study.

We have included this point in a future study.

Underweight is highly prevalent in Japanese inpatients with schizophrenia. Psychiatrists should be aware of underweight and their potential health risks. Treating psychiatrists should also be responsible for providing the necessary nutritional interventions [45]. Future research should focus on evaluating interventions that target underweight as well as obesity. And, the main study should include an analysis of differences by age and differences from healthy participants to examine variations in muscle mass.

Under 2. Materials and Methods- Please indicate the study design, i.e., case-control study.

Response: Thank you for your advice. We have revised the sentence of 2.1. Study Participants.

This pilot case-control study enrolled 60 individuals in total comprising 30 healthy participants (healthy group) and 30 patients with schizophrenia (patient group), matched by age, ranging from 40 to 89 years, and sex.

Under 3. Results, lines 163 to 175, it would be better if differences between male and female (or by sex) were also presented since study participants were matched by age and sex (lines 85 and 86), or at least include it in the table. For age, I see it was used as a continuous variable. Do you think it would be good to categorize age at least into two categories (adults and older adults (65 years old and above) to examine variations in muscle mass?

Response: The points reviewer mentioned were not conducted in this pilot study due to the small number of participants. However, we believe that in future main studies, they need to be categorized and analyzed according to the categories you suggested. We have added that point to our future research. Thank you very much.

Future research should focus on evaluating interventions that target underweight as well as obesity. 

The association of low body water and dynapenia in patient participants suggests that low body water might be a risk factor for dynapenia in these patients. Furthermore, the main study following this pilot study should include an analysis of differences by age and differences from healthy participants to examine variations in low body water, visceral fat level, and muscle mass.

Lines 157-162 states that 10% had sarcopenia (sounds diagnostic, conclusive). How about stating that 10% met the criteria for sarcopenia since no physical performance was measured as part of the assessment of sarcopenia?

Response: Thank you for your advice. We have revised it.

In this study, dynapenia and sarcopenia/presarcopenia were assessed. Among the 30 participants of the patient group, 10.0% (3/30 [1/18 women, 2/12 men]) met the criteria for sarcopenia, 3.3% (1/30 [1/12 men]) presarcopenia, and 60.0% (18/30 [14/18 women, 4/12 men]) dynapenia. The corresponding results of the healthy group showed that sarcopenia and presarcopenia were not present (0%), 13.3% (4/30 [3/19 women, 1/11 men]) met the criteria for dynapenia. Table 1 shows the results of Welch's t-test.

Reviewer 2 Report

1. How is a healthy group defined? Since those in the Health Group are employees and are still working, will this not have problems when compared with those who are admitted as patients?

2. Why is the height not obtained? Why is it taken from records or through self-report?

3. The discussion should explain how the disorder contributes to the loss of muscles.

4. The texts written in the discussion are to be transferred to the findings to allow enough space for the explanation of what has been found.

Author Response

Comments and Suggestions for Authors

Response: Thanks for your advice and suggestive comments. Revisions are shown in red font.

  1. How is a healthy group defined? Since those in the Health Group are employees and are still working, will this not have problems when compared with those who are admitted as patients?

Response: A healthy group defined as a control group of persons without schizophrenia. Employees over 65 years of age work in the hospital in their good health, we considered no problem in comparison with patients. This case-controlled study was also been approved by the hospital's Ethics Review Committee.

  1. Why is the height not obtained? Why is it taken from records or through self-report?

Response: The text has been revised as follows because the hospital measures height during the hospital's employee physical examinations.

Healthy group: Age and height were self-reported based on the hospital's staff health examination form

  1. The discussion should explain how the disorder contributes to the loss of muscles.

Response: In the discussion section, we discussed how disability contributes to muscle loss considering previous research.

Appetite regulation and physical activity affect energy balance and changes in body fat mass. In some patients, inflammation induces anorexia and fat loss along with sarcopenia. In others, appetite is maintained, despite the activation of systemic inflammation, leading to sarcopenia with normal or increased BMI. Inactivity contributes to sarcopenia and increased fat tissue in aging and disease [31].

In the previous study of the BMI status of hospitalized Japanese schizophrenia patients, underweight and obesity were characteristic in schizophrenia inpatients compared with the general population. In particular, regarding the characteristics of underweight, a previous study showed that the prevalence of hypotriglyceridemia was significantly higher in the underweight group than in the normal weight group and in overweight/obese schizophrenia inpatients [32].

  1. The texts written in the discussion are to be transferred to the findings to allow enough space for the explanation of what has been found.

Response: We have summarized the results of this study in the first paragraph of the discussion and edited out redundancies. Thank you very much.

Reviewer 3 Report

Examining Risk Factors for Dynapenia/Sarcopenia in Patients with Schizophrenia: A Pilot Study

Tanioka et. al., Healthcare

The authors present an examination of the prevalence of muscle wasting in patients with schizophrenia compared to healthy individuals.  The conclusions of this study are concise, straightforward, and in agreement with the results.  The study is of general interest to the field because it reveals muscle wasting as an important comorbidity within schizophrenic patients.  The article is well written and, as a reviewer, I see little preventing it from being published in its present form.  Several minor, but important considerations are listed below.  

Introduction

Line 61-62: Sedentary lifestyle is not “caused by patients with schizophrenia”. Please rephrase.

Materials and Methods

There should be some sort of an IRB approval statement within the methods section     

Results

The results suggest a healthy BMI for both groups, but show a higher Body fat percentage for patients with schizophrenia. These were all inpatients, so this needs to be reconciled with the statement cited in the introduction, line 75, reference 15, indicating a high prevalence of an underweight presentation in Japanese inpatients with schizophrenia.

Please ensure that the Tables are not split between two pages.  Based upon the size of the tables, it is possible to fit them on 1 page, and this should be done for ease of interpretation.

Discussion

Referring to the comment about underweight schizophrenic inpatients above, what was the general composition of the diet for the patients in this study and how might it have compared to the healthy group?  How does this diet compare or contrast to the diet of patients in other studies, such as those in reference 15?  These concepts are alluded to in lines 248-263, but previous studies in which the diet compositions are well known should be discussed in a paragraph within the discussion

Author Response

Comments and Suggestions for Authors

The authors present an examination of the prevalence of muscle wasting in patients with schizophrenia compared to healthy individuals. The conclusions of this study are concise, straightforward, and in agreement with the results. The study is of general interest to the field because it reveals muscle wasting as an important comorbidity within schizophrenic patients.  The article is well written and, as a reviewer, I see little preventing it from being published in its present form. Several minor, but important considerations are listed below. 

Response: Thanks for your advice and suggestive comments. Revisions are shown in red font.

Introduction

Line 61-62: Sedentary lifestyle is not “caused by patients with schizophrenia”. Please rephrase.

Response: Thanks for your advice. The following modifications have been made and references were added as follows. Also, introduction section was thoroughly reviewed for language, grammar and tone. Some paragraph breaks were added for clarity.

In the context of socioeconomic challenges, schizophrenia leads to particularly unhealthy lifestyles that include poor diets, little exercise, marked sedentary behavior, and high rates of smoking with commensurately low physical activity levels [12]. As a result, compared to the general healthy population, patients with schizophrenia have severe symptomatic limitations in physical capacity. Negative symptoms reduce the likelihood of patients’ engagement in goal-directed behavior, including physical activity, which has been noted to increase obesity and cardiometabolic risk and induce poor physical condition, resulting in sarcopenic obesity and muscle weakness [5].

Materials and Methods

There should be some sort of an IRB approval statement within the methods section.    

Response: The format of this journal allows it to be placed after the Conclusion.

Results

The results suggest a healthy BMI for both groups, but show a higher Body fat percentage for patients with schizophrenia. These were all inpatients, so this needs to be reconciled with the statement cited in the introduction, line 75, reference 15, indicating a high prevalence of an underweight presentation in Japanese inpatients with schizophrenia.

Response: We have included this point in a future study.

Underweight is highly prevalent in Japanese inpatients with schizophrenia. Psychiatrists should be aware of underweight and their potential health risks. Treating psychiatrists should also be responsible for providing any necessary nutritional interventions [45]. Future research should focus on evaluating interventions that target underweight as well as obesity.

The association of low body water and dynapenia in patient participants suggests that low body water might be a risk factor for dynapenia in these patients. Furthermore, the main study following this pilot study should include an analysis of differences by age and differences from healthy participants to examine variations in low body water, visceral fat, and muscle mass.

Please ensure that the Tables are not split between two pages. Based upon the size of the tables, it is possible to fit them on 1 page, and this should be done for ease of interpretation.

Response: Tables 1 and 2 are arranged by one page.

Discussion

Referring to the comment about underweight schizophrenic inpatients above, what was the general composition of the diet for the patients in this study and how might it have compared to the healthy group?

Response: Visceral fat level (t = 2.11, p = 0.04), body fat (t = 2.76, p <0.01), and BMI (t = 2.11, p = 0.04) were significantly higher in the patient group than in the healthy group. Thus, visceral fat level, body fat, and BMI were significantly higher in the patient group than in the healthy group. However, the average value of the patient group in BMI is not at the obese level, and the high visceral fat level was considered as a problem when considered overall from the cross tabulation results in Table 2.

Therefore, we have revised the discussion sentences like below:

As shown in Table 1, the patient group had significantly higher body fat, visceral fat level scores, and BMI. However, it was suggested that being afflicted with schizophrenia is one factor associated with dynapenia. Also, the average value of the patient group BMI is not at the obese level, and the high visceral fat level score was deemed a problem when considered overall from the cross-tabulation results in Table 2.

How does this diet compare or contrast to the diet of patients in other studies, such as those in reference 15? These concepts are alluded to in lines 248-263, but previous studies in which the diet compositions are well known should be discussed in a paragraph within the discussion.

Response: The following sentences have were added to the last part of the discussion section.

Based on the BMI findings of our study, the patient group was not underweight. Our study subjects were inpatients; they have consumed a diet regulated by a psychiatrist and a dietitian. However, outpatients may not be eating an appropriate diet due to unbalanced diets, poverty, etc. With this in mind, we should conduct the main case-control study following this pilot study.

Again, we appreciate your appropriate advice.